# The Perceived Health Status from Young Adults to Elderly: Results of the MEHM Questionnaire within the CUORE Project Survey 2008–2012

**DOI:** 10.3390/ijerph17176160

**Published:** 2020-08-25

**Authors:** Claudia Giacomozzi, Luigi Palmieri, Lidia Gargiulo, Cinzia Lo Noce, Laura Iannucci, Anna Di Lonardo, Serena Vannucchi, Graziano Onder, Furio Colivicchi, Simona Giampaoli, Chiara Donfrancesco

**Affiliations:** 1Department of Cardiovascular and Endocrine-Metabolic Diseases and Aging, National Institute of Health, 00161 Rome, Italy; luigi.palmieri@iss.it (L.P.); cinzia.lonoce@iss.it (C.L.N.); anna.dilonardo@iss.it (A.D.L.); serena.vannucchi@iss.it (S.V.); graziano.onder@iss.it (G.O.); simonagiampaoli@outlook.com (S.G.); chiara.donfrancesco@iss.it (C.D.); 2Department for Statistical Production—Directorate for Social Statistics and Welfare, Italian National Statistical Institute, 00100 Rome, Italy; gargiulo@istat.it (L.G.); iannucci@istat.it (L.I.); 3National Association Hospital Cardiologists—Health Care Foundation, 50121 Florence, Italy; furio.colivicchi@aslroma1.it; 4U.O.C. Cardiologia Clinica e Riabilitativa, Presidio Ospedaliero San Filippo Neri—ASL Roma 1, 00135 Rome, Italy

**Keywords:** perceived health status, health behaviors, prevention, health examination survey

## Abstract

Improving healthy life years requires an effective understanding and management of the process of healthy ageing. Assessing the perceived health status and its determinants is a relevant step in this process. This study explored the potentialities of the Minimum European Health Module (MEHM) to cope with this critical issue. Investigation was conducted on 4798 Italian residents (49.7% women, aged 35–79 years), participating in the CUORE Project Health Examination Survey 2008–2012. The three MEHM questions—perceived health status, chronic morbidity and activity limitations—were examined also in association with living context, seasonality, marital status and level of education. A higher prevalence of health status negative perception was associated with older age (9% and 24% respectively in men and women aged 35–44 years; 46% and 61% respectively in men and women aged 75–79 years). In women, this negative perception was higher than in men in any age group, and reached 50% in the 65–69 age group, 10 years earlier than in men. For both sexes, the level of education had a strong impact on this negative perception (odds ratio 2.32 and 2.72 in men and women respectively), while “living alone” played a greater impact in women than in men. MEHM activity limitations subscale was as much as 30% higher for questionnaires answered during the hottest months. This study identified potential predictors of perceived health status in adults aged 35–79 years, which can be used to target interventions aimed at improving self-perceived health status.

## 1. Introduction

The ageing of the world population is a growing phenomenon, and has more than doubled in the last 200 years, with total life expectancy increasing by five years between 2000 and 2015 globally. A slower age-related deterioration of cognitive and motor functions has been observed as well, which is the core concept of the so-called “healthy ageing”. However, healthy life years, namely that part of life free from diseases, are yet too much shorter than lifespan itself, with up to 16–20% of late-life morbidity. Research on individuals living extremely long lives, and on the specific association of physical activity with the risk of frailty in healthy older adults [1] suggested that factors like the environment, lifestyle, socio-economic status and incremental factors such as life-long diet, education and physical activity all together play relevant roles not only on the life span itself, but also on the perceived health status, which is a milestone of the overall healthy ageing process [2]. Reasonably, a deeper understanding of the role and impact of these factors can help compress morbidity, especially in cases of type II Diabetes, cardiovascular diseases, obesity, chronic lumbar and musculoskeletal pain, thus improving perceived health status and, consequently, healthy life years.

Information gathered through the Minimum European Health Module (MEHM, [3]) seems to support the exploration of the active and healthy ageing dimension through the quantification of perceived health status [4,5]. This Module consists of a set of three general questions characterizing three different concepts of health, namely self-perceived health status at a very general level, chronic morbidity, and long-standing activity limitations due to health problems. The module (link: http://ec.europa.eu/eurostat), developed in the late 1990s, to be used in social surveys, is currently included in the European Health Interview Survey (EHIS) and EU Statistics on Income and Living Conditions (EU-SILC) [6,7].

Each of these three, apparently simple questions has a strong rationale behind. The self-perceived health assessment item (Item 1), which is based on the WHO recommendations [8], is associated with different dimensions of health, i.e., physical, social and emotional function, and biomedical signs and symptoms. A strong predictor of future functional limitations, cognitive impairment and mortality [9,10,11], Item 1 seems to be suitable to make a comparison between different populations [12]. The chronic morbidity item (Item 2), developed by the Italian National Institute of Statistics [13], relies on the strong correlation between long-standing diseases and worsening of health-related quality of life. Finally, the long-standing activity limitations (Item 3, known as the Global Activity Limitations Indicator (GALI)) aim at identifying those individuals who perceive themselves as having long-standing, health-related restrictions or limitations in their usual activities. This latter item, highly predictive of functional problems associated with activity restrictions, appears to be especially relevant when investigating ageing populations, and is often used to calculate the healthy life years indicator and to account for health expenditure [14,15,16,17,18].

The reliability of each of the three MEHM questions was considered acceptable, with slightly higher values for men than for women [3].

This study examined data from the MEHM survey collected during the Osservatorio Epidemiologico Cardiovascolare/Health Examination Survey (OEC/HES) 2008–2012 within the framework of the CUORE Project [19,20], with the following aims: (1) to describe and explore age-related and sex-related differences in the answers to the three MEHM questions by the Italian general population aged 35–79 years; (2) to investigate associations of perceived health status with some relevant demographic factors—context of living, marital status, education—and with seasonality, namely the season in which the MEHM questionnaire was administered and answered. This first descriptive phase of the study can help acquire new knowledge and better model predictors of health status deterioration, and represents the preliminary analysis for a second phase on the assessment of the association between MEHM and measured and self-reported health status.

## 2. Materials and Methods

### 2.1. Data Extraction and Pre-Processing

In the 2008–2012 period, as part of the CUORE Project, the Istituto Superiore di Sanità/Italian National Institute of Health (ISS) in collaboration with the Associazione Nazionale Cardiologi Ospedalieri/National Association of Hospital Cardiologists (ANMCO) and the Heart Care Foundation (HCF) conducted the OEC/HES on random samples of the general Italian population aged 35–79 years resident in all Italian Regions [19,20]. A sample of 220 people per 1.5 million inhabitants was selected, thus guaranteeing at least one sample for each region, even those with a smaller population. The samples were selected at random from the register of residents aged 35–79 years, in order to recruit 25 men and 25 women for each 10-year age group between 35–74 years (35–44, 45–54, 55–64, 65–74) and 10 men and 10 women in the age group of 75–79 years. People were invited to register by mail letter; participants were informed about the aims of the research by means of a project information note and in this way each participant was able to sign and give his/her informed consent to participate. To assess the participation rate, the following categories of people were considered ineligible and removed from the original sample: the dead, emigrants, those working outside the area of residence for the entire survey period, and those whose undelivered letters were returned with the notation “unknown”.

The OEC/HES 2008–2012 was approved by the Ethical Committee of the ISS on 11 November 2009 and was recognized as part of the Joint Action of the European Health Examination Survey (EHES) [21].

Appropriate training of the survey staff [21] ensured homogeneity in the presentation and administration of the questions to all participants. The self-administered questionnaire, provided by the National Institute of Statistics [22] in the version in force in 2008, included questions related to some of the internationally validated questionnaires, including activities of daily life and instrumental activities of daily life [23,24,25,26], and the Italian validated version of MEHM [3,27]. The OEC/HES 2008–2012 was implemented in all Italian Regions, but the MEHM questionnaire was administered in 11 regions (out of 20) of Northern, Central and Southern Italy (Valle d’Aosta, Liguria, Lombardy, Trentino Alto Adige, Veneto, Marche, Tuscany, Umbria, Abruzzo, Campania, Puglia). More details on the study design and the sample size of OEC/HES 2008–2012 are reported elsewhere [28].

Within these 11 regions, the participation rate, defined as the number of people who participated in the survey after receiving the invitation divided by the size of the eligible sample, was 57% [19,20,29].

Participants were instructed to return the completed MEHM forms before leaving the examination room; if necessary, they could count on the support of the staff.

The answers to the three MEHM questions collected and considered in the analysis were:Q1 (MEHM Item 1, general health status). The question “How is your health in general?” had five possible answer categories: very good; good; fair; bad; and very bad. In order to comply with the requirement of homogeneity of data processing, the order of the 5 answers was inverted to obtain a score of increasing rather than decreasing satisfaction, i.e., very bad perception of health status; bad perception of health status; neither good nor bad; good; very good;Q2 (MEHM Item 2, chronic morbidity). The question “Do you have any long-standing illness or health problem?” had two possible answers: yes (long-standing compromised health status) or no (uncompromised health status); with respect to the meaning of “long-standing”, it was explained that it should be understood as a period of not less than 6 months;Q3 (MEHM Item 3, activity limitations). The question “For at least the past 6 months, to what extent have you been limited because of a health problem in activities people usually do? Would you say you have been …” had three possible answers: severely limited; limited but not severely; not limited at all. The answers were organized on the basis of an increasing satisfaction score.

All responses relating to one MEHM item were grouped into one ‘negative’ and one ‘positive’ response, based on the following pooling criteria: Q1 was scored as negative either for “very bad”, or “bad”, or “neither bad nor good”, and as positive for either “good” or “very good”; Q2 was scored as negative for “yes” (long-standing compromised health status) and as positive for “no” (uncompromised health status); Q3 was scored as negative for either “serious limitations” or “minor limitations”, and as positive for “no limitations”.

The following environmental and demographic factors were also collected and assessed during the OEC/HES 2008–2012:The context in which the participant lived, classified as rural and urban, the latter applicable to municipalities with 10,000 or more inhabitants [30,31];The season, referring to the month in which the questionnaire was administered and codified from 1 January to 12 December;The marital status of the participant collected through a face-to-face questionnaire, classified as single, never married; married or cohabiting; separated or divorced; widow/widower;The participant’s level of education collected through a face-to-face questionnaire, classified as bachelor’s degree, master’s degree or PhD; high school diploma; lower secondary school licence; primary school licence; level of education lower than primary school.

For the purposes of this analysis, the factors examined were dichotomized as follows: based on the usual temperatures in Italy, months were assigned to seasons as follows: months from October to March were associated with cold season, months from April to September were associated with hot season; marital status was coded as married (married or cohabitant status) or single (single, divorced, widowed); the level of education was coded as high (degree/PhD or high school diploma) or low (remaining lower levels of education); the context remained unchanged.

### 2.2. Data Processing and Statistical Analysis

Preliminary data analysis aimed at verifying data reliability and consistency. Cronbach’s Alpha reliability test was thus applied to the MEHM items; this technique is recommended for self-administered psychometric questionnaires, with high reliability guaranteed by values ≥ 0.70 [32]. The test proved a good consistency of the MEHM answers: median was 0.70 [0.64–0.71] over the 18 groups for the raw MEHM answers, and 0.68 [0.65–0.70] for the dichotomized (negative/positive) answers (also compliant with all 2-test requirements). Corresponding median correlation was 0.44 [0.37–0.45] (Pearson’s correlation coefficients among the MEHM items, within each group).

For the analyses, men and women were stratified by age groups of 5 years, from 35–39 to 75–79 years. Answers to the three MEHM items were assessed either to explore possible mutual influence among themselves—especially the influence of Q2 and Q3 on Q1, analyzed separately for each age group—or to investigate possible relationships between MEHM items and confounding factors (irrespectively of age group).

Analyses were performed separately for men and women. Frequencies and their 95% confidence intervals (CI) were assessed. For each dichotomized MEHM item, percentages of negative answers were calculated as the ratio between the number of negative answers and the total number of answers within each group of interest.

Comparisons between age groups were done using 2 tests, and adjusted *p*-values for multiple comparisons were calculated according to the Bonferroni correction. Each class of age was initially compared with the 35–39 years class until a comparison resulted statistically significant; following age classes were compared with the age class corresponding with the statistically significant comparison, and so on.

A logistic regression analysis was conducted to explore the relationships of each dichotomized item—Q1, Q2 and Q3—with age classes (reference age class 35–39 years) and environmental and demographic factors, as well as to explore the association among the three items. Odd ratios and their 95% CIs were obtained by applying a logistic regression model to each dichotomized MEHM item (dependent variable) as age class function and environmental and demographic factors (independent variables), and to Q1 (dependent variable) as a function of the other two items Q2 and Q3 (independent variables) adjusted for age class and environmental and demographic factors. In this preliminary and exploratory analysis, all factors have been included individually in order to better understand and evaluate the contribution of each factor net of all other factors, which have however been taken into account.

The “best” (least impacting) and the “worst” (most impacting) combination of environmental and demographic factors was calculated as the lowest and the highest percentage of simultaneously negative Q1, Q2 and Q3 answers.

Data processing and statistical analysis were performed through MatlabR2015a (The MathWorks, Inc., Natick, MA, USA), OriginPro8 (OriginLab Corporation, Northampton, MA, USA) and R3.5.2 (R Foundation, Vienna, Austria).

## 3. Results

### 3.1. Analysis of Age and Gender Groups

A total of 4798 complete MEHM questionnaires (2387 from women and 2411 from men) were available and included in the analysis, over a total of 4865 persons (2427 women and 2438 men) examined in the 11 regions (percentage of non-returned MEHM questionnaires: 1.4%). Participants were equally distributed between the two sex groups. Although the group of older women was—as expected due to the age classes sample sizes assumption—slightly smaller than the others, the age group distribution of women was not statistically different from that of men (2 test, *p* = 0.181). The 18 groups (5-years each, 9 for women and 9 for men), ranged from a minimum of 181 to a maximum of 320 participants (non-uniform distribution, Kolmogorov–Smirnov test *p* < 0.0001). In each group, median age corresponded to the mean age, with 25–75° range of 3 years (with only two minor exceptions). Detailed data can be found in Table A1 of Appendix A.

As expected, the perception of general health status, chronic morbidity and activity limitations was worse in older groups, with negative scores ranging from 9–24% in the two youngest groups (men and women respectively) and reaching 46–63% in the two oldest groups (men and women respectively).

It should also be noted that:women showed a worse perception of their health status than men at any age (negative perception of general health status (Q1) ranging from 19 to 62% for women and 9 to 46% for men) and there was a greater slope from younger to older age (43% and 37% for women and 37% for men, respectively); as a result, half of the women interviewed (50%) reported a negative perceived health status at the age of 65–69, i.e., ten years earlier than men (Figure 1);a relevant prevalence of chronic morbidity (Q2: 24 and 16% in women and men respectively) and activity limitations (Q3: 17 and 15% in women and men respectively) was found even among young adults (35–39 years old), which is quite remarkable in the general population;for both genders, 8 multiple comparisons were required to detect statistically significant differences of all MEHM items by age groups. Specifically, a first significant difference was found between the 35–39 group and the 50–54 group, and a second one between the 50–54 and the 65–69 groups; a third one was finally found between the 65–69 group and the oldest group, but only for general health status and activity limitations (Figure 1);differences were almost linearly distributed with respect to age groups in both women and men.

For all MEHM items, logistic regression models showed that poor health perception was significantly higher for both men and women aged 50 and over than for younger groups (Table 1). All three items were found to be associated with the level of education, and activity limitations were found to be associated with all environmental and demographic factors (Table 1). The perceived health status was statistically associated with chronic morbidity and activity limitations even when adjusted for age, environmental and demographic factors (Table 1).

The frequency distributions of MEHM answers were detailed in Figure A1 (original and complete answers) and Figure A2 (dichotomous answers).

### 3.2. Global Analysis on Environmental and Demographic Factors

The potential impact of context, season, marital status and level of education on MEHM dichotomized outcomes was investigated in two groups, women and men, without any age stratification. For each factor, frequency distributions showed no statistically significant differences (2-test, *p* > 0.05) between women and men. For both gender groups, the level of education was the factor that had the greatest impact, with a perceived negative health condition up to 50% in the case of a low level of education (Figure 2). The negative perception of health conditions appeared slightly higher in the urban context; however, this factor needs to be interpreted carefully, as only 14% of the participants lived in a rural context. The season in which the MEHM questionnaire was administered had a significant impact on women, with a perception of activity limitations up to 12% higher in the warmer/hotter months. Women also showed a worse perception of health condition when living alone, especially in terms of activity limitations.

Seasonality, marital status and level of education were then pooled and analyzed in any of their possible combinations and results were summarized in Figure 3. Based on the percentage of simultaneously negative answers to all MEHM items (in Figure 3, represented by the dotted bar for combined items Q1 and Q2 and Q3), the best perceived health condition (lowest percentage), both for women and men, resulted from cohabitation or marriage, combined with high level of education and MEHM administration during the cold season (percentages were 11% and 10% respectively). For men, the worst perceived health condition (highest percentage) was associated with cohabitation or marriage, low level of education, and MEHM administration during the hot season (17%). For women, the worst perceived health condition was associated with being single, low level of education, and MEHM administration during the hot season (25%). This combination, however, was only found in 7% of participants, thus its outcome deserves additional investigation.

## 4. Discussion

The standardized, quality-controlled administration of MEHM questionnaires may represent a valuable tool to investigate differences in health status perception, burden of chronic diseases and activity limitations on cohorts of healthy individuals. In the present study, it allowed to assess and model these health perception indicators for both genders, starting from early adult age. In Italy, in the 35–79 age group, the prevalence of perceived negative health status ranged from 19 to 61% for women and from 9 to 46% for men; the prevalence of chronic diseases ranged from 24 to 52% for women and from 16 to 50% for men; the prevalence of activity limitations ranged from 17 to 63% for women and from 15 to 49% for men. A not negligible prevalence of a perceived negative health status in young adults may apparently be surprising; however, it is well in agreement with the paradigm of human physical and cognitive decline. As summarized in the 2018 review on the global challenges of ageing [2], in fact, the ageing process begins from the age of thirty, albeit at a subclinical level. Initial changes in the composition and function of bones, cartilage and muscles are often followed by an increase in abdominal fat, changes in the endocrine system, blood pressure and blood lipids, insulin resistance, mechanical and structural changes that can affect heart and brain functions. In middle age, diseases can become clinically defined, often giving rise to a multi-morbidity scenario. In the elderly, among whom many studies have shown that self-perceived health status is a significant predictor of mortality [33], frailty can further complicate the clinical status, adding severe functional and cognitive limitations.

Thus, while research is increasingly deepening genetic studies on human life span variation, identification and validation of biomarkers of the physiological state and biological age of individuals [2], MEHM-based investigation of very large cohorts of individuals can provide useful information on life course health-related trends and perceptions in association with changes in the world scenario—be they social, environmental, demographic or political.

As evidenced by the data in this study, although the MEHM items are, as expected, strongly correlated with each other, with a strong dependence of the health status perception on long-standing diseases and activity limitations, there are some important aspects that are better highlighted if each of the three items is properly investigated as a standalone indicator. Within each gender group, in fact, the prevalence of negative answers almost linearly correlates with age, but with peculiar slopes and offsets for each item (Figure 1): negative health perception and activity limitations in women show a comparable rate (42% and 46% respectively), much higher than the chronicity rate (28%); the three items have a comparable rate in men (37%, 34% and 36% respectively), where chronicity and activity limitations result in smaller differences of perceived health status until very old age (>75 years). Although the third MEHM item was addressed as an inclusive tool, with only one question, good and sufficient predictive and simultaneous validity and reliability [17], the results of this study are corroborated by studies that investigated the different impact of chronic conditions and activity limitations on health care expenditure: while conceptually related, they present different dimensions of ill-health, and the resulting information can be genuinely complementary [18]. In this study, while for men there seems to be partial disagreement with the position expressed above, as they show a parallel trend but a lower prevalence of activity limitations related to long-standing diseases, in women these two factors—chronic condition and activity limitations—appear only weakly correlated, and from the 65–69 age group, functional limitations have a greater impact than chronicity.

Based on all three items, a very important aspect observed is that women have a generally worse perception of their health status than men throughout their adult lives. In both women and men, in fact, this is almost linearly correlated with age, but it shows a higher negative prevalence in women than in men in the youngest adult group (35–39 years), and a higher slope (Figure 1). This brings to an earlier manifestation of a critical “changing point”, namely the age group in correspondence of which half of the participants perceive a negative health status: in women, in fact, this critical point occurs in the 65–69 age group, while in men it only appears ten years later, in the 75–79 age group (Figure 1). This finding is consistent with some studies investigating the active and healthy ageing (AHA) dimension focused on healthy life years (HLY) expectancy; in particular, a paradox, called the gender-related health-survival paradox, characterizes the European population: while women’s mortality advantage contributes to more HLY in women than in men, the higher prevalence of disability in women reduces the HLY difference between genders [5]. This longer late-life morbidity in women than in men was reaffirmed in qualified recent literature [2]. Interestingly, data from the present study showed that women have higher prevalence of negative scores in all three MEHM items for each age-matched comparison; this suggests that more attention should be paid to this phenomenon and that possible preventive actions should start very early in adult life.

The analysis of the living context (urban or rural), seasonality (cold or hot season), marital status (married/cohabitant or single) and level of education (high or low) showed that health condition appeared slightly worse in the urban context, more so when the questionnaire was administered during the hot season and in the case of single marital status (especially in the group of women), and appeared significantly worse in the case of low level of education. However, the role of the living context should be investigated in more detail, as only 14% of the whole population enrolled could be referred to a rural context. By pooling the other three factors and analyzing every possible combination, very interesting observations could be made: for both women and men, the best health perception corresponded to a married/cohabiting condition, combined with a high level of education, but only when the questionnaire was completed during the cold season; while the worst health condition was associated with a married/cohabiting condition, a low level of education and MEHM questionnaire delivery during the hot season. Actually, the most negative health condition for women was associated with being single (all other conditions unchanged); however, only 7% of the participants fell into this sub-group with this specific combination, which therefore deserves to be the subject of future investigations. The negative impact of hot season, though partly unexpected, is quite reasonable in Italy, where the hot temperatures reached in summer can make health conditions and disease-related limitations more difficult to cope with, especially for older citizens living in an urban context. Loneliness, too, may have a major impact under these conditions, and is worthy of attention in future studies. Any additional age-dependent survey will certainly help to better understand these issues.

This preliminary secondary investigation on the possible roles of some social, demographic and environmental factors proved to be in line with the thorough and valuable work of Marmot et al. [34], which deals with the WHO European Review of social determinants of health and the health divide. In accordance with the review, and without considering accurate social and economic factors—which are likely to be included in a broader analysis of the data—it could nevertheless be observed that gender, as well as environmental conditions, isolation and education have an impact on the general perception of health status and the burden of chronic diseases and activity limitations. In particular, the findings of this study, which show a slightly better perception of health status in a rural context rather than in an urban context, need to be further investigated: Marmot, in fact, reported a similar trend in association with poor economic conditions that forced people to live in partially degraded and unhealthy urban areas. However, it is possible that, despite economic reasons, living a long life in an urban area may lead to an increase in another important health determinant, namely the level of stress.

### Strengths and Limitations

The main strengths of this study are the following: the good national coverage with the enrolment of study participants through random stratification by age and gender in more than half of the Italian Regions distributed in Northern, Central and Southern Italy; a wide age range and a satisfactory participation rate; the use of a validated questionnaire with a very limited percentage of non-returned questionnaires.

On the other hand, some study limitations are hereby acknowledged, which should be taken into account when interpreting the findings of this study, among which:-unbalanced number of urban and rural samples;-easonality; in each region, the data collection took place in the same season, but the latitude of the various regions is different, which can potentially influence the association of MEHM with seasonality; however, the distributions of questionnaires in hot and cold seasons was quite balanced for both women and men (51% vs. 49%), moreover the geographical areas were fairly homogeneously represented for the hot season, and also for the cold season if we consider that in Italy the cold season is quite similar in the central and southern regions;-different living conditions, experience, expectations in the oldest and youngest groups—including the effect of the age cohort—which may have affected comparisons;-the subgroups provided by urban/rural, married/single, high/low level of education do not always have a similar distribution in terms of age and gender, as these characteristics could not be known in advance;-the cross-sectional design of the study does not allow to assess the causality of the associations between factors;-the lack of information on the accessibility of health care facilities may have influenced the perceived health status, even though this aspect could be partly included in the level of education;-potential bias may have derived from the design of the study, which required participants to “come to the examination room”. Non-respondents may have had various reasons for refusing, among which mobility limitations due to health condition, feeling of a healthy or “already under control” status, opposite feeling of being “too sick or too fearful to attend a screening”, or simply lack of availability for the screening for practical/working reasons; furthermore the perceived relevance or sensitivity of the topic may affect participation.

## 5. Conclusions

These findings are the result of a survey conducted on a wide range of Italian adult population, including early adults, grouped according to a rather narrow age-range criterion (5 years). The very limited percentage of non-returned questionnaires confirmed the feasibility of administering MEHM survey at any age. The study findings suggested that trends, perception and signs of health deterioration shall be investigated and monitored through a wide life span starting from young adult age. The 5-year grouping helped to identify significant differences at the ages of 50–54, 65–69 and 75–79. Assessing the three MEHM items while simultaneously accounting for possible confounding effects of age, environmental and demographic factors highlighted a major role of the level of education and marital status. Seasonality also seemed to have an impact on the perception of chronic diseases burden and on the overall health status perception. The results of this survey can be a useful contribution to the planning of timely preventive actions or comprehensive public health strategies to address the multidimensional process of ageing.

## Figures and Tables

**Figure 1 ijerph-17-06160-f001:**
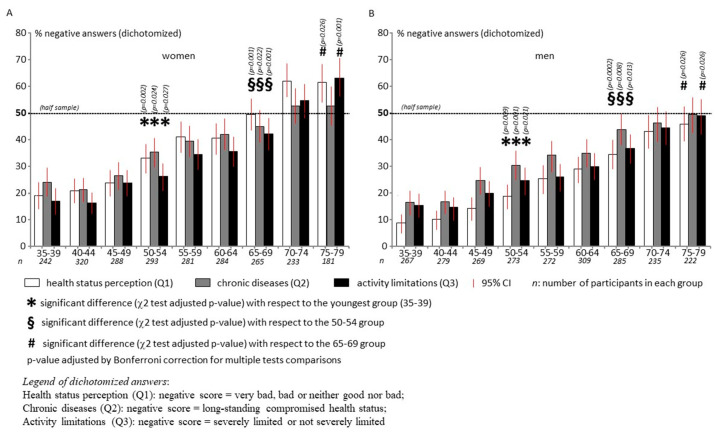
Prevalence and 95% confidence intervals (CIs) of negative answers to the dichotomized the Minimum European Health Module (MEHM) items, plotted for women (**A**) and for men (**B**) (data source: the CUORE Project Health Examination Survey 2008–2012, Italian men and women, 35–79 years old).

**Figure 2 ijerph-17-06160-f002:**
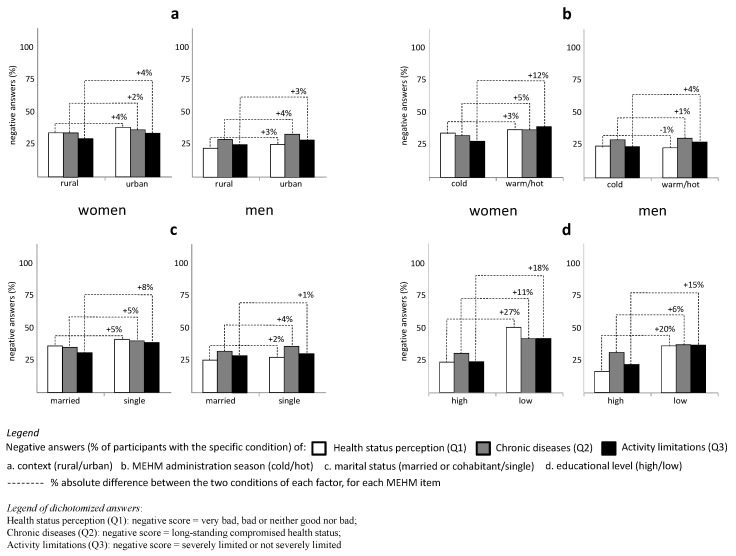
Negative score (%) of MEHM items in association with living context (**a**), seasonality (namely, the season in which the MEHM questionnaire was administered) (**b**), marital status (**c**) and level of education (**d**), each factor accounted individually and separately plotted for men and for women. Plots are referred to dichotomized variables (data source: the CUORE Project Health Examination Survey 2008–2012, Italian men and women, 35–79 years old).

**Figure 3 ijerph-17-06160-f003:**
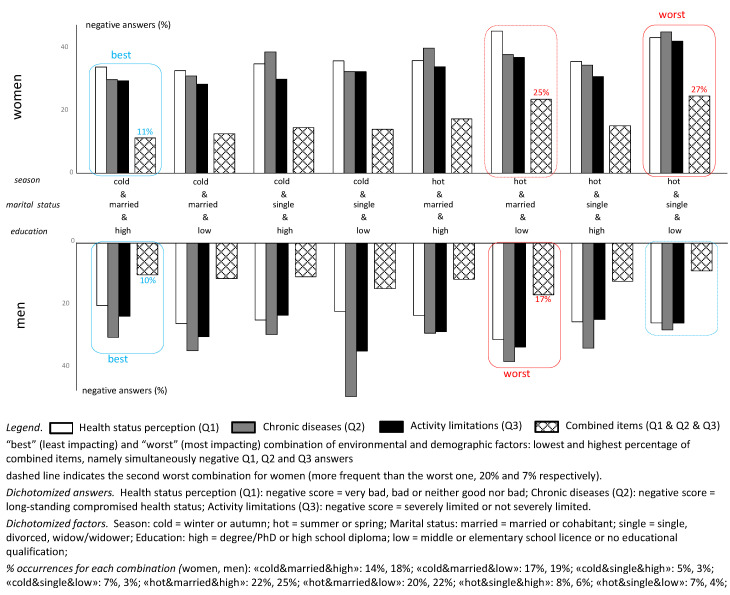
Negative score (%) of MEHM items in any combination of conditions such as seasonality (cold/hot), marital status (married/single) and level of education (high/low), distinguishing between men and women. Plots are referred to dichotomized variables (data source: the CUORE Project Health Examination Survey 2008–2012, Italian men and women, 35–79 years old). Combinations of factors were identified as worst or best on the basis of the percentage—highest and lowest, respectively—of participants who, for each specific combination of factors, gave simultaneous negative answers to all MEHM items.

**Table 1 ijerph-17-06160-t001:** Logistic regression models of dichotomized Minimum European Health Module (MEHM) (data source: the CUORE Project Health Examination Survey 2008–2012, Italian men and women, 35–79 years old).

			Intercept	Age 40–44	Age 45–49	Age 50–54	Age 55–59	Age 60–64	Age 65–69	Age 70–74	Age 75–79	Q2	Q3	Education	Marital Status	Season	Context
General health status (Q1) ^a^	W	OR	0.14 ***	1.12	1.21	2.03 ***	2.52 ***	2.24 ***	3.06 ***	4.46 ***	4.42 ***			2.29 ***	1.14	1.15	1.13
	W	95% CI	0.09; 0.22	0.74; 1.72	0.79; 1.87	1.35; 3.08	1.68; 3.81	1.49; 3.39	2.03; 4.65	2.91; 6.93	2.81; 7.06			1.89; 2.77	0.92; 1.39	0.96; 1.39	0.87; 1.47
	M	OR	0.07 ***	1.11	1.68	2.23 **	3.27 ***	3.73 ***	4.51 ***	6.70 ***	6.84 ***			1.76 ***	1.20	1.00	1.16
	M	95% CI	0.04; 0.11	0.62; 2.00	0.98; 2.93	1.33; 3.82	1.99; 5.52	2.31; 6.22	2.78; 7.55	4.11; 11.28	4.17; 11.59			1.43; 2.17	0.92; 1.57	0.82; 1.22	0.87; 1.56
Chronic morbidity (Q2) ^a^	W	OR	0.22 ***	0.87	1.12	1.72 **	1.93 ***	2.16 ***	2.39 ***	2.91 ***	2.85 ***			1.25 *	1.13	1.37 *	1.10
	W	95% CI	0.15; 0.32	0.58; 1.30	0.75; 1.67	1.18; 2.53	1.32; 2.85	1.48; 3.19	1.62; 3.55	1.94; 4.39	1.85; 4.42			1.03; 1.51	0.92; 1.37	1.15; 1.64	0.86; 1.42
	M	OR	0.14 ***	1.06	1.69 *	2.36 ***	2.85 ***	2.89 ***	4.39 ***	4.76 ***	5.36 ***			0.93	1.36 *	1.16	1.23
	M	95% CI	0.09; 0.21	0.67; 1.67	1.10; 2.60	1.56; 3.59	1.90; 4.34	1.94; 4.35	2.94; 6.67	3.13; 7.31	3.50; 8.32			0.77; 1.12	1.07; 1.73	0.97; 1.38	0.95; 1.61
Activity limitations (Q3) ^a^	W	OR	0.11 ***	0.97	1.48	1.74 *	2.35 ***	2.46 ***	3.14 ***	4.40 ***	6.43 ***			1.55 ***	1.24 *	1.55 ***	1.21
	W	95% CI	0.07; 0.17	0.62; 1.53	0.96; 2.30	1.14; 2.68	1.55; 3.62	1.62; 3.79	2.06; 4.85	2.85; 6.89	4.04; 10.39			1.27; 1.88	1.01; 1.52	1.29; 1.87	0.92; 1.58
	M	OR	0.11 ***	0.93	1.35	1.79 **	1.89 **	2.19 ***	2.96 ***	4.08 ***	4.38 ***			1.41 ***	1.06	1.32 **	1.20
	M	95% CI	0.07; 0.18	0.58; 1.50	0.86; 2.14	1.17; 2.79	1.23; 2.93	1.45; 3.36	1.95; 4.55	2.67; 6.34	2.84; 6.86			1.16; 1.71	0.82; 1.37	1.09; 1.60	0.91; 1.58
General health status (Q1) ^b^	W	OR	0.12 ***									2.69 ***	5.17 ***	2.72 ***	1.13	0.94	1.07
	W	95% CI	0.08; 0.17									2.17; 3.33	4.15; 6.44	2.22; 3.33	0.90; 1.41	0.77; 1.15	0.80; 1.44
	M	OR	0.08 ***									3.42 ***	4.24 ***	2.32 ***	1.01	0.89	1.01
	M	95% CI	0.05; 0.11									2.70; 4.32	3.36; 5.37	1.87; 2.90	0.75; 1.34	0.72; 1.11	0.74; 1.39
General health status (Q1) ^c^	W	OR	0.09 ***	1.19	1.06	1.71 *	1.94 **	1.60 *	2.08 **	2.75 ***	2.31 **	2.58 ***	4.77 ***	2.26 ***	1.05	0.93	1.07
	W	95% CI	0.05; 0.14	0.75; 1.90	0.66; 1.71	1.09; 2.71	1.24; 3.07	1.02; 2.52	1.31; 3.32	1.71; 4.48	1.39; 3.88	2.07; 3.21	3.82; 5.98	1.82; 2.80	0.83; 1.31	0.76; 1.14	0.80; 1.44
	M	OR	0.04 ***	1.14	1.49	1.74	2.54 ***	2.86 ***	2.88 ***	4.07 ***	3.94 ***	3.07 ***	3.99 ***	1.44 ***	1.12	0.88	1.03
	M	95% CI	0.02; 0.07	0.61; 2.13	0.83; 2.70	1.00; 3.09	1.49; 4.47	1.70; 4.95	1.71; 5.00	2.38; 7.13	2.30; 6.94	2.42; 3.89	3.15; 5.06	1.84; 2.32	0.83; 1.51	0.70; 1.10	0.75; 1.46

Odds ratio (OR) and 95% confidence interval (CI) for OR of the logistic regression models for women (W) and men (M) (reference: age class 35–39 years): ^a^ each dichotomized Minimum European Health Module (MEHM) item (Q1, Q2 or Q3: dependent variable) as a function of age and of education, marital status, season and context (independent variables); ^b^ Q1 item “general health status” (dependent variable) as a function of the independent variables Q2 “chronic morbidity”, Q3 “activity limitations”, education, marital status, season and context (simple model); ^c^ Q1 (dependent variable) as a function of the independent variables Age, Q2, Q3, education, marital status, season and context (age-adjusted model). Legend for the statistical significance of the model coefficient: *** *p*-value < 0.001; ** *p*-value < 0.01; * *p*-value < 0.05. The models explained small amount of variance, with estimates ranging 0.04–0.09 for individual items (“a” models) and reaching 0.23 (men) and 0.24 (women) for generalized models including the three items (“c” models). Health status perception (Q1): negative score = very bad, bad or neither good nor bad; positive score = good or very good; chronic diseases (Q2): negative score = long-standing compromised health status; positive score = uncompromised health status; activity limitations (Q3): negative score = severely limited or not severely limited; positive score = not limited at all. Education: high = degree/PhD or high school diploma; low = middle or elementary school license or no educational qualification; marital status: married = married or cohabitant; single = single, divorced, widow/widower; season: cold = winter or autumn; hot = summer or spring; context: urban or rural.

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
