# Peer review of "The Perceived Health Status from Young Adults to Elderly: Results of the MEHM Questionnaire within the CUORE Project Survey 2008–2012"

_ijerph, 2020, doi:10.3390/ijerph17176160_

Round 1

Reviewer 1 Report

This is a well written and interesting manuscript, however concerns about the methodology limit enthusiasm for publication.

First, the article lacks clarity in its objective, which may explain the methodological problems, but there is also a lack of relationship with the conclusions. While this is a cross-sectional design and the authors indicate that they aim to describe and explore differences in the perceived health status by demographic, environmental, social and educational factors, they conclude with the recommendation to plan preventative actions addressing the multi-dimension process of ageing, the optimization of ergonomics, solutions for health maintenance and rehabilitation, conceived to be fruited in either home, work or recreational settings. Either of those recommendations could be suggested with a cross-sectional design with the data they analyze.

A major lack of this work is "Table 1" where the main socio-demographic and health characteristics of the sample are shown. This is critical to understand the associations found and shown in this work. it is very important to understand what is the distribution of the different factors here analyzed to make sure that there is a balanced distribution, for instance, of those who answered the survey in hot or cold months. This aspect is important: because it´s difficult to believe that responding on hot or cold months may be associated with a higher prevalence of chronic conditions, as appears in women in this work. It is also important to clarify whether the hot-cold period is equally distributed across participating regions: without being an expert in Italian geography it is highly likely the existence of significative weather differences between regions as Aosta or Trentino and Campania or Puglia.

The cumulative negative answers that authors show in Figure 2 and Figure 3 are either clearly aligned with an objective to "describe and explore differences in the perceived health status". The logic of aggregating negative answers is really complex to understand, taking also into account that the specific approach to how those percentages are obtained is not mentioned in the Methods sections. It may be interesting to develop an "index" based on perceived health status with or not negative answers to the other questions, but as they are shown now are not a valid measure.

objective of 

Author Response

Global feedback to comments.

We are grateful to the Editor and the Reviewers for the careful review of our manuscript; it allowed us to better address the aims of the study and to improve the manuscript quality and readability. All comments and suggestions have been addressed, and detailed explanations are hereby reported point-by-point. Two supplemental Tables (A,2 and A,3) have been removed and one supplemental Table (A,2) has been added. The three Figures have been properly edited. Two references have been added. The manuscript has been analysed for typos and language check. To cope with all comments, we needed to slightly increase the length of the manuscript.

REVIEWER 1

This is a well written and interesting manuscript, however concerns about the methodology limit enthusiasm for publication.

First, the article lacks clarity in its objective, which may explain the methodological problems, but there is also a lack of relationship with the conclusions. While this is a cross-sectional design and the authors indicate that they aim to describe and explore differences in the perceived health status by demographic, environmental, social and educational factors, they conclude with the recommendation to plan preventative actions addressing the multi-dimension process of ageing, the optimization of ergonomics, solutions for health maintenance and rehabilitation, conceived to be fruited in either home, work or recreational settings. Either of those recommendations could be suggested with a cross-sectional design with the data they analyze.

Thank you for the suggestion. Changes have been reported in the Introduction (last paragraph reviewed) and in the Conclusions (whole section reviewed)

Introduction: the last paragraph has been modified as follows: “This study examined data from the MEHM survey collected during the Osservatorio Epidemiologico Cardiovascolare/Health Examination Survey (OEC/HES) 2008-2012 within the framework of the CUORE Project ([19-20]), with the following aims: 1) to describe and explore age-related and sex-related differences in the answers to the three MEHM questions by the Italian general population aged 35-79 years; 2) to investigate associations of perceived health status with some relevant demographic factors – context of living, marital status, education – and with seasonality, namely the season in which the MEHM questionnaire was administered and answered. This first descriptive phase of the study can help acquire new knowledge and better model predictors of health status deterioration, and represents the preliminary analysis for a second phase on the assessment of the association between MEHM and measured and self-reported health status

Conclusions: the whole section has been modified as follows: “These findings are the result of a survey conducted on a wide range of Italian adult population, including early adults, grouped according to a rather narrow age-range criterion (5 years). The very limited percentage of non-returned questionnaires confirmed the feasibility of administering MEHM survey at any age. The study findings suggested that trends, perception and signs of health deterioration shall be investigated and monitored through a wide life span starting from young adult age. The 5-year grouping helped to identify significant differences at the ages of 50-54, 65-69 and 75-79. Assessing the three MEHM items while simultaneously accounting for possible confounding effects of age, environmental and demographic factors highlighted a major role of the level of education and marital status. Seasonality also seemed to have an impact on the perception of chronic diseases burden and on the overall health status perception. The results of this survey can be a useful contribution to the planning of timely preventive actions or comprehensive public health strategies to address the multidimensional process of ageing.”

A major lack of this work is "Table 1" where the main socio-demographic and health characteristics of the sample are shown. This is critical to understand the associations found and shown in this work. it is very important to understand what is the distribution of the different factors here analyzed to make sure that there is a balanced distribution, for instance, of those who answered the survey in hot or cold months. This aspect is important: because it´s difficult to believe that responding on hot or cold months may be associated with a higher prevalence of chronic conditions, as appears in women in this work. It is also important to clarify whether the hot-cold period is equally distributed across participating regions: without being an expert in Italian geography it is highly likely the existence of significative weather differences between regions as Aosta or Trentino and Campania or Puglia.

Thank you for the suggestion. The issue has been addressed by adding the following Table (new Table A,2) in the Supplemental material

Table A,2 NEW

Number of answers as distributed with respect to the two groups of participants (women and men) and the two conditions for each factor.

Distributions are detailed with respect to the geographical position (North, Center or South of Italy) of the involved regions.

(data source: the CUORE Project health examination survey 2008-2012, Italian men and women, 35-79 years old)

Geographical area of Italy

factor

WOMEN

MEN

factor

WOMEN

MEN

URBAN

RURAL

northern

866

870

218

218

central

310

327

118

116

southern

875

880

0

0

total

2051(86%)

2077(86%)

336(14%)

334(14%)

HOT

COLD

northern

423

430

661

658

central

428

443

0

0

southern

373

355

502

525

total

1224(51%)

1228(51%)

1163(49%)

1183(49%)

HIGH ED

LOW ED

northern

528

568

556

520

central

226

220

202

223

southern

420

463

455

417

total

1174(49%)

1251(52%)

1213(51%)

1160(48%)

MARRIED

SINGLE

northern

751

879

333

209

central

304

351

124

92

southern

682

779

193

101

total

1737(73%)

2009(83%)

650(27%)

402(17%)

Legend. Percentages are calculated with respect to the total of the returned MEHM questionnaires for women and men separately

Regions within each geographical area of Italy: Northern = Liguria, Lombardy, Trentino Alto Adige, Valle d’Aosta, Veneto; Central = Marche, Tuscany, Umbria; Southern = Abruzzo, Campania, Puglia.

ED: education

For the specific issue of seasonality factor, distributions were quite balanced both for women and for men (51% vs 49% for both genders). Geographical areas were quite homogeneously represented for the hot season, and considering that cold season of central and southern Regions was quite similar in Italy, also for the cold season geographical areas were quite homogeneously represented.

However, the issue has been accounted for in the new Strengths and Limitations section. Specifically, we mentioned that “in each region, the data collection took place in the same season, but the latitude of the various regions is different, which can potentially influence the association of MEHM with seasonality; however, the distributions of questionnaires in hot and cold seasons was quite balanced for both women and men (51% vs 49%), moreover the geographical areas were fairly homogeneously represented for the hot season, and also for the cold season if we consider that in Italy the cold season is quite similar in the central and southern regions

Finally, distribution of all confounding factors has been commented in the Strengths and Limitations as follows:

On the other hand, some study limitations are hereby acknowledged, which should be taken into account when interpreting the findings of this study, among which:

-    unbalanced number of urban and rural samples;

-    seasonality; in each region, the data collection took place in the same season, but the latitude of the various regions is different, which can potentially influence the association of MEHM with seasonality; however, the distributions of questionnaires in hot and cold seasons was quite balanced for both women and men (51% vs 49%), moreover the geographical areas were fairly homogeneously represented for the hot season, and also for the cold season if we consider that in Italy the cold season is quite similar in the central and southern regions;

-    different living conditions, experience, expectations in oldest and youngest groups – the effect of the age cohort –may have affected comparisons;

-    the subgroups provided by urban/rural, married/single, high/low level of education do not always have a similar distribution in terms of age and gender, as these characteristics could not be known in advance;

-    the cross-sectional design of the study does not allow to assess the causality of the associations between factors;

-    the lack of information on the accessibility of health care facilities may have influenced the perceived health status, even though this aspect could be partly included in the level of education.”

The cumulative negative answers that authors show in Figure 2 and Figure 3 are either clearly aligned with an objective to "describe and explore differences in the perceived health status". The logic of aggregating negative answers is really complex to understand, taking also into account that the specific approach to how those percentages are obtained is not mentioned in the Methods sections.

It may be interesting to develop an "index" based on perceived health status with or not negative answers to the other questions, but as they are shown now are not a valid measure.

Thank you for the comment. To address this issue:

  • the confounding term “cumulative” has been removed: actually, the percentual negative answers of the three MEHM items had been simply stacked one over the other so as to improve graphical readability;
  • in both Figures (2 and 3), relevant details have been added to the vertical axes so as to allow an easier interpretation of the reported data;
  • Methods section has been reviewed to better clarify how percentages of negative answers had been calculated; specifically:
    • the MEHM items scores have been defined as “negative” or “positive” rather than “1” or “0” as in the previous version of the manuscript;
    • the following explanation has been added: “For each dichotomized MEHM item, percentages of negative answers were calculated as the ratio between the number of negative answers and the total number of answers within each group of interest. Comparisons between age groups were done using c2 tests, and adjusted p-values for multiple comparisons were calculated according to the Bonferroni correction. The initial reference class was 35-39 years; it was replaced from time to time with the age group for which there was statistical significance by comparing this one with older groups.”
  • about Figure 3,
    • the explanation of the method used to define the best and worst combination of MEHM items is now reported in the text (Data processing and statistical analysis paragraph) and in the footnote of the Figure: the “best” (least impacting) and the “worst” (most impacting) combination of environmental and demographic factors had been calculated as the lowest and the highest percentage of simultaneously negative Q1, Q2 and Q3 answers.
    • the combined items percentages for the best and the worst combinations (11% and 27% respectively for women; 10% and 17% respectively for men) have been added to the plot
    • in the attempt to further improve readability, in all figures (Figure 1 included) white, grey and black colours have been used for the Q1, Q2 and Q3 bars instead of the previously used textures.

Reviewer 2 Report

The paper „The perceived health Status from young adults to elderly: results of the MEHM questionnaire within CUORE Project survey 2008-2012” is based on data collected in CUORE Project in different Italian regions and aims on assessment of health status measured in three main domains: self-perceived health, presence of chronic diseases and activity limitations.  The problem is interesting, however this paper needs some improvements.

First, this paper is based on the data from 11 out of 16 Italian regions, then the sample size for all age-gender groups is similar – so some additional explanation how random sample was achieved and what was inclusion criteria should be added to this paper and later discussed. Also in regards to response rate (57%?) .

I have also some remarks to the environmental and behavioral factors – first they are mainly demographic characteristics: context (place of living), marital status, education level – I have not found any “behavioral” factors here.

The next comments are related to the presentation of the analysis results.

  1. The non-response rate is not 1.4% - this is just percentage of not-returned MEHM questionnaires (or not completed questionnaires).
  2. “participants were uniformly distributed among age and sex groups” – that is not true – I have tested if these 18 groups have uniform distribution and p-value in this test is <0.001. The p-value presented by authors in the bottom of Tab. A1 (p=0,181) is for Chi2 test comparing distribution of age groups between males and females, not for equal number of respondents in each of 18 age-sex groups.
  3. I would suggest to move the results of Cronbach’s alpha and correlation analysis from description of the results to the Material and methods section. In addition add the information about what kind of correlation (Pearson’s, Spearman’s, etc.) and the variables for which this correlation coefficients were estimated.
  4. Do not use terms lite “increase / decrease / worsening” to describe the differences between age (or sex) groups. This study has cross-sectional design – so each respondent has filled-in questionnaire only once. Use terminology like – higher / lower prevalence / rare / frequency.
  5. Do not use “amount of prevalence” – prevalence is just a frequency measure. In addition possible effect of the age cohort should be discusses, especially between the oldest and the youngest groups – like different living conditions, experience, expectations, etc.
  6. For the comparison of prevalence of “negative answers” in age groups (vs the youngest one) there is a problem of multiple comparisons – so adjust p-values (and add the information about method of adjustment in Statistical analysis description) for multiple comparisons.
  7. There is no need to repeat exactly the same data both in table and on figure (Tab A.2 and Fig. A.1 as well as tab A.3 and fig. A.2). In table A.3 there is no clear information about the meaning of specific categories Q…_1 and Q…_2 – below the table one can find explanation about codes: 0 and 1.
  8. It table A.2 the data are presented as percentage while in table A.3 the same type of data is presented as decimals.
  9. The data in tab. 3 are not presented properly – in this table OR is presented (=exp(b)) together with 95%CI but for b not OR. It should be presented as OR and 95%CI for OR.
  10. The presentation of prevalence of Q1, Q2 and Q3 as cumulative data is not a proper way of data presentation – these variables are not mutually exclusive – the person who declared for example presence of chronic disease, can have also activity limitations as well as low self-perceived health.  On figures 2 and 3 there is no labels for Y axis and clear comparison between presented groups is not so easy.
  11. The description of the limitations as well as the discussion of their possible impact on results of this study should be added.
  12. The English should be carefully checked.

Author Response

Global feedback to comments.

We are grateful to the Editor and the Reviewers for the careful review of our manuscript; it allowed us to better address the aims of the study and to improve the manuscript quality and readability. All comments and suggestions have been addressed, and detailed explanations are hereby reported point-by-point. Two supplemental Tables (A,2 and A,3) have been removed and one supplemental Table (A,2) has been added. The three Figures have been properly edited. Two references have been added. The manuscript has been analysed for typos and language check. To cope with all comments, we needed to slightly increase the length of the manuscript.

Detailed point-by-point Reply to Reviewer 2 are delivered in the attached file

Reviewer 3 Report

Dear Claudia Giacomozzi,

I attach the recommendations for your article.

Best regards,

Author Response

Global feedback to comments.

We are grateful to the Editor and the Reviewers for the careful review of our manuscript; it allowed us to better address the aims of the study and to improve the manuscript quality and readability. All comments and suggestions have been addressed, and detailed explanations are hereby reported point-by-point. Two supplemental Tables (A,2 and A,3) have been removed and one supplemental Table (A,2) has been added. The three Figures have been properly edited. Two references have been added. The manuscript has been analysed for typos and language check. To cope with all comments, we needed to slightly increase the length of the manuscript.

REVIEWER 3

This is an article focus on of an area with good justification and need to go deeper into the topic The perceived health status from young adults to 2 elderly: results of the MEHM questionnaire within 3 the CUORE Project survey 2008-2012. Overall most methods have been employed to a good standard and described well.

I have a few comments and suggestions to help improve clarity in parts of the paper:

In the abstract: perhaps it would be interesting to clearly specify the objective of this study.

Thank you for the comment. Within the word limits of the Abstract, we tried to better specify the study main aim. The Abstract has been modified as follows: “Improving healthy life years requires an effective understanding and management of the process of healthy ageing. Assessing the perceived health status and its determinants is a relevant step in this process. This study explored the potentialities of the Minimum European Health Module (MEHM) to cope with this critical issue. Investigation was conducted on 4798 Italian residents (49.7% women, aged 35-79 years), participating in the CUORE Project health examination survey 2008-2012. The three MEHM questions - perceived health status, chronic morbidity and activity limitations - were examined also in association with living context, seasonality, marital status and level of education. Descriptive analysis and logistic regressions showed that a higher prevalence of health status negative perception was associated with older age; in women, this negative perception was higher than in men in any age group, and reached 50% in the 65-69 age group, 10 years earlier than in men. For both sexes, the level of education had a strong impact on MEHM answers, while “living alone” played a greater impact in women than in men. MEHM activity limitations subscale was as much as 30% higher for questionnaires answered during the hottest months. This study identified potential predictors of perceived health status in young and older adults. These results can be used to target interventions aimed at improving self-perceived health status.”

In the introduction, line 41: it would perhaps be good to mention that there are previous systematic reviews that analyze physical activity, for example: Physical activity and the risk of frailty among community-dwelling healthy older adults: A protocol for systematic review and meta-analysis. Bei Pan, Hongli Li, Yunhua Wang, Min Sun, Hui Cai, and Jiancheng Wang, Medicine (Baltimore). 2019 Aug; 98(35): e16955. doi: 10.1097/MD.0000000000016955.

Thank you for the comment. The reference has been added to References and briefly cited in the Introduction.

Line 43-44: perhaps it would be interesting to include diseases associated with the aging process and lifestyles such as type II diabetes, cardiovascular disease and premature mortality, for example overweight-obesity and low back discomfort, effect on musculoskeletal symptoms.

Thank you for the comment. The sentence has been completed as follows: “Reasonably, a deeper understanding of the role and impact of these factors can help compress morbidity, especially in cases of type II Diabetes, cardiovascular diseases, obesity, chronic lumbar and musculoskeletal pain, thus improving perceived health status and, consequently, healthy life years.

Lines 129-133: It would be interesting to specify the objective of the study..

Thank you for the suggestion. Changes have been reported in the Introduction (last paragraph reviewed) and in the Conclusions (whole section reviewed)

Introduction: the last paragraph has been modified as follows: “This study examined data from the MEHM survey collected during the Osservatorio Epidemiologico Cardiovascolare/Health Examination Survey (OEC/HES) 2008-2012 within the framework of the CUORE Project ([19-20]), with the following aims: 1) to describe and explore age-related and sex-related differences in the answers to the three MEHM questions by the Italian general population aged 35-79 years; 2) to investigate associations of perceived health status with some relevant demographic factors – context of living, marital status, education – and with seasonality, namely the season in which the MEHM questionnaire was administered and answered. This first descriptive phase of the study can help acquire new knowledge and better model predictors of health status deterioration, and represents the preliminary analysis for a second phase on the assessment of the association between MEHM and measured and self-reported health status

Conclusions: the whole section has been modified as follows: “These findings are the result of a survey conducted on a wide range of Italian adult population, including early adults, grouped according to a rather narrow age-range criterion (5 years). The very limited percentage of non-returned questionnaires confirmed the feasibility of administering MEHM survey at any age. The study findings suggested that trends, perception and signs of health deterioration shall be investigated and monitored through a wide life span starting from young adult age. The 5-year grouping helped to identify significant differences at the ages of 50-54, 65-69 and 75-79. Assessing the three MEHM items while simultaneously accounting for possible confounding effects of age, environmental and demographic factors highlighted a major role of the level of education and marital status. Seasonality also seemed to have an impact on the perception of chronic diseases burden and on the overall health status perception. The results of this survey can be a useful contribution to the planning of timely preventive actions or comprehensive public health strategies to address the multidimensional process of ageing.”

Material and methods: How were the participants selected? What type of sampling? It would be interesting to include this information

Thank you for the suggestion. Methodological details have been added to address the raised issue. Specifically, two paragraphs and two references have been added to the Methods section, and one paragraph modified as follows:

A sample of 220 people per 1.5 million inhabitants was selected, thus guaranteeing at least one sample for each region, even those with a smaller population. The samples were selected at random by gender from the register of residents aged 35-79 years, in order to recruit at least 50 individuals for each 10-year age group between 35-74 years (35-44, 45-54, 55-64, 65-74) and 20 individuals in the age group of 75-79 years

To assess the participation rate, the following categories of people were considered ineligible and removed from the original sample: the dead, emigrants, those working outside the area of residence for the entire survey period, and those whose undelivered letters were returned with the notation "unknown". The OEC/HES 2008-2012 was approved by the Ethical Committee of the ISS on 11 November 2009 and was recognized as part of the Joint Action of the European Health Examination Survey (EHES). [Tolonen H, EHES Manual: Part B. Fieldwork procedures, 2016. Directions 2016_14. Available at http://www.julkari.fi/handle/10024/131503]

Within these 11 Regions, the participation rate, defined as the number of people who participated in the survey after receiving the invitation divided by the size of the eligible sample, was 57% [Mindell JS, Giampaoli S, Goesswald A, et al. Sample selection, recruitment and participation rates in health examination surveys in Europe--experience from seven national surveys. BMC Med Res Methodol. 2015;15:78. Published 2015 Oct 5. doi:10.1186/s12874-015-0072-4]

Line 299: “…in our study”…perhaps it is better to write “in the present study or in the current study or in this study” Line 311: “…our finding”…perhaps it is better to write “in the findings of this study”. Line 341: the same “our study”. Please review it throughout the text.

Thank you for the comment. Whole manuscript has been checked to remove any reference to “our” study or findings

Conclusions: Perhaps it would be interesting to include the strengths and weaknesses of the present study.

Thank you for the suggestion. The issue has been addressed in the Strengths and Limitations section:

The main strengths of this study are the following: the good national coverage with the enrolment of study participants through random stratification by age and gender in more than half of the Italian Regions distributed in Northern, Central and Southern Italy; a wide age range and a satisfactory participation rate; the use of a validated questionnaire with a very limited percentage of non-returned questionnaires.

 On the other hand, some study limitations are hereby acknowledged, which should be taken into account when interpreting the findings of this study, among which:

-    unbalanced number of urban and rural samples;

-    seasonality; in each region, the data collection took place in the same season, but the latitude of the various regions is different, which can potentially influence the association of MEHM with seasonality; however, the distributions of questionnaires in hot and cold seasons was quite balanced for both women and men (51% vs 49%), moreover the geographical areas were fairly homogeneously represented for the hot season, and also for the cold season if we consider that in Italy the cold season is quite similar in the central and southern regions;

-    different living conditions, experience, expectations in oldest and youngest groups – the effect of the age cohort –may have affected comparisons;

-    the subgroups provided by urban/rural, married/single, high/low level of education do not always have a similar distribution in terms of age and gender, as these characteristics could not be known in advance;

-    the cross-sectional design of the study does not allow to assess the causality of the associations between factors;

-           the lack of information on the accessibility of health care facilities may have influenced the perceived health status, even though this aspect could be partly included in the level of education.”

Reviewer 4 Report

This study identifies potential predictors of perceived health status with a structural questionnaire in young and older adults. I raise my comments as below to the authors.

1.The participation rate was 57%. Can the authors provide a comparison of sex-aged composition between the groups participated or not?

2.What’s the results of Cronbach's Alpha reliability test?

3.Table 1 showed the OR and its 95% CI. That could make some confusions for readers since they presented not so matched. The range of CI should include its OR. Moreover, lots of OR for environmental/behavior factors are more than one. A reference group for each environmental/behavior factor should be indicated to represent the opposite group having a higher odds?

4.The study stated that this factor has to be interpreted carefully since only 13% of participants lived in a rural context. However, the rate of 14% was showed in Fig 2.

A Y-axial scale of Fig 2 is suggested.

5.Seasonality, marital status and educational level were pooled and analyzed. Since the three factors have a strong correlation with each other, the combination for analysis needs a reason.

6.The study concluded that health worsening seems to increase linearly with age. I think the statement can be true if the authors used a cohort study.

7.Health condition perception may be affected by accessibility of health care facility. The authors are suggested to raise some confounders as well as study limitations.

8.Line 361 “Spam” can be replaced with “span”.

9.Table A.2 and Fig A.1 have the same presentation and are suggested to remain one. So do Table A.3 and Fig A.2.

10.Lines 473-483 are no sense.

Author Response

Global feedback to comments.

We are grateful to the Editor and the Reviewers for the careful review of our manuscript; it allowed us to better address the aims of the study and to improve the manuscript quality and readability. All comments and suggestions have been addressed, and detailed explanations are hereby reported point-by-point. Two supplemental Tables (A,2 and A,3) have been removed and one supplemental Table (A,2) has been added. The three Figures have been properly edited. Two references have been added. The manuscript has been analysed for typos and language check. To cope with all comments, we needed to slightly increase the length of the manuscript.

REVIEWER 4

This study identifies potential predictors of perceived health status with a structural questionnaire in young and older adults. I raise my comments as below to the authors.

The participation rate was 57%. Can the authors provide a comparison of sex-aged composition between the groups participated or not?

Thank you for the comment.

A reference has been added with more details by gender and age in relation to the participation rate for the overall 20 regions (Mindell JS, Giampaoli S, Goesswald A, et al. Sample selection, recruitment and participation rates in health examination surveys in Europe--experience from seven national surveys. BMC Med Res Methodol. 2015;15:78. Published 2015 Oct 5. doi:10.1186/s12874-015-0072-4).

Methodological details have been added to in the Methods section as follows:

A sample of 220 people per 1.5 million inhabitants was selected, thus guaranteeing at least one sample for each region, even those with a smaller population. The samples were selected at random by gender from the register of residents aged 35-79 years, in order to recruit at least 50 individuals for each 10-year age group between 35-74 years (35-44, 45-54, 55-64, 65-74) and 20 individuals in the age group of 75-79 years

To assess the participation rate, the following categories of people were considered ineligible and removed from the original sample: the dead, emigrants, those working outside the area of residence for the entire survey period, and those whose undelivered letters were returned with the notation "unknown". The OEC/HES 2008-2012 was approved by the Ethical Committee of the ISS on 11 November 2009 and was recognized as part of the Joint Action of the European Health Examination Survey (EHES). [Tolonen H, EHES Manual: Part B. Fieldwork procedures, 2016. Directions 2016_14. Available at http://www.julkari.fi/handle/10024/131503]

Within these 11 Regions, the participation rate, defined as the number of people who participated in the survey after receiving the invitation divided by the size of the eligible sample, was 57% [Mindell JS, Giampaoli S, Goesswald A, et al. Sample selection, recruitment and participation rates in health examination surveys in Europe--experience from seven national surveys. BMC Med Res Methodol. 2015;15:78. Published 2015 Oct 5. doi:10.1186/s12874-015-0072-4]

What’s the results of Cronbach's Alpha reliability test?

Thank you for the comment.

To better clarify results, aim and results of the analysis have been better explained.

The reviewed period is now as follows: “Preliminary data analysis aimed at verifying data reliability and consistency. Cronbach's Alpha reliability test was thus applied to the MEHM items; this technique is recommended for self-administered psychometric questionnaires, with high reliability guaranteed by a values ≥ 0.70 [31]. The test proved a good consistency of the MEHM answers: median a was 0.70 [0.64-0.71] over the 18 groups for the raw MEHM answers, and 0.68 [0.65-0.70] for the dichotomized (negative/positive) answers (also compliant with all c2-test requirements). Corresponding median correlation was 0.44 [0.37-0.45] (Pearson’s correlation coefficients among the MEHM items, within each group)

Table 1 showed the OR and its 95% CI. That could make some confusions for readers since they presented not so matched. The range of CI should include its OR. Moreover, lots of OR for environmental/behavior factors are more than one. A reference group for each environmental/behavior factor should be indicated to represent the opposite group having a higher odds?

Thank you for the comment. We replaced them with the 95%CI for OR. Further, the reference group, namely the youngest age group, was clearly stated both in the text and in the Table 1 legend.

The study stated that this factor has to be interpreted carefully since only 13% of participants lived in a rural context. However, the rate of 14% was showed in Fig 2.

Thank you for the comment. We apologize for the misprint, which has been corrected. The correct value is 14% as shown in the Figure.

A Y-axial scale of Fig 2 is suggested.

Thank you for the comment. . A Y-axial scale was added to Fig 2 as suggested.

Seasonality, marital status and educational level were pooled and analyzed. Since the three factors have a strong correlation with each other, the combination for analysis needs a reason.

Thank you for the comment. Despite the expected positive correlation between marital status and educational level, we found that in our cohort the slope was quite different between women and men (WOMEN: married & high ed: 36%; married & low ed: 37%; single & high ed: 13%; single & low ed: 14%; MEN: married & high ed: 42%; married & low ed: 41%; single & high ed: 9%; single & low ed: 7%). Further, no association might be expected between these two factors and the season when the MEHM questionnaire had been administered. Thus, in this preliminary, exploratory analysis we proposed to evaluate the contribution of each factor while accounting for the others.

We added a short explanatory sentence to the Methods section as follows: “In this preliminary and exploratory analysis, all factors have been included individually in order to better understand and evaluate the contribution of each factor net of all other factors, which have however been taken into account

The study concluded that health worsening seems to increase linearly with age. I think the statement can be true if the authors used a cohort study.

Thank you for the comment. We checked the whole manuscript to replace any occurrence of terms not suitable for cross-sectional studies (like increase/decrease/worsening..), and added the following comment to the Strengths and Limitations section “the cross-sectional design of the study does not allow to assess the causality of the associations between factors

Health condition perception may be affected by accessibility of health care facility. The authors are suggested to raise some confounders as well as study limitations.

Thank you for the comment. A new Strengths and Limitations section has been added:

The main strengths of this study are the following: the good national coverage with the enrolment of study participants through random stratification by age and gender in more than half of the Italian Regions distributed in Northern, Central and Southern Italy; a wide age range and a satisfactory participation rate; the use of a validated questionnaire with a very limited percentage of non-returned questionnaires.

 On the other hand, some study limitations are hereby acknowledged, which should be taken into account when interpreting the findings of this study, among which:

-    unbalanced number of urban and rural samples;

-    seasonality; in each region, the data collection took place in the same season, but the latitude of the various regions is different, which can potentially influence the association of MEHM with seasonality; however, the distributions of questionnaires in hot and cold seasons was quite balanced for both women and men (51% vs 49%), moreover the geographical areas were fairly homogeneously represented for the hot season, and also for the cold season if we consider that in Italy the cold season is quite similar in the central and southern regions;

-    different living conditions, experience, expectations in oldest and youngest groups – the effect of the age cohort –may have affected comparisons;

-    the subgroups provided by urban/rural, married/single, high/low level of education do not always have a similar distribution in terms of age and gender, as these characteristics could not be known in advance;

-    the cross-sectional design of the study does not allow to assess the causality of the associations between factors;

-    the lack of information on the accessibility of health care facilities may have influenced the perceived health status, even though this aspect could be partly included in the level of education

Line 361 “Spam” can be replaced with “span”.

Thank you for the comment. We apologize for the misprint, which has been corrected.

Table A.2 and Fig A.1 have the same presentation and are suggested to remain one. So do Table A.3 and Fig A.2.

Thank you for the comment. We thus removed Tables A.2 and A.3, since we thought Figures might support a more immediate interpretation of results.

Lines 473-483 are no sense.

Thank you for the comment. We apologize for the misprint, which has been removed.

Round 2

Reviewer 1 Report

This version of the manuscript included significant changes that clarify the analysis and is now suitable for publication.

Author Response

We thank you the Reviewer for the positive feedback.

Reviewer 2 Report

Authors introduced some changes to their paper, but not all changes (important ones) have been introduced and the paper still need to be improved.

  1. Abstract – some “meaningful” results should be displayed numerically – for example general information as “”a greater impact in women than in men” is meaningless –the strength of this impact should be displayed.
  2. Abstract – I have not found any information about “predictors” in “young and older adults” – this kind of analysis was performed in the whole group. The only analysis in age groups was the assessment of distributions of MEHM items.
  3. Add the information about the total sample size required for this study (or just information what is the size of the population aged 35-79 years) in all selected regions (for this analysis). It allows the reader to assess the possibility of making any conclusion about the population.
  4. I understand that participants were invited to come to “examination room” – so it can make the study sample more “healthy” that general population, especially in the oldest groups.
  5. Add the information which months were assigned to cold and hot seasons – instead of information about the seasons of the year.
  6. The “interactions” among MEHM items were not assessed.
  7. For the X2 test (with correction for multiple comparisons) there is no need to give the information about the “reference category” – in this analysis there is no reference category – the analysis is made for comparisons of all possible pairs of categories.
  8. I do not understand why only 8 multiple comparisons were performed to compare 9 age groups? – To choice the “reference category” one needs to perform each – by each comparison.
  9. The lower number of participants from the oldest group is obvious because of the assumption for the sample size made (only 20 per 1.5 milion).
  10. Figure 1 – it is not clear for which of three analyzed items the difference exists between index age group and this indicated by (*, $, #). Below the table exact p-values should be displayed rather than p<0.05.
  11. The results of logistic regression should not be interpreted as probability because “the odds ratio” are calculated.
  12. I have still concerns about the data as well as interpretation of data presented on figures 2 and 3. Because there are three items displayed on figure 2 and all these items are not mutually exclusive they should not be displayed as cumulative bar!!!. They cab be displayed as three separate bars. Figure 3 – it data are for “simultaneously negative Q1, Q2 and Q3 answers” – it should be presented as one bar per any of combinations of season x marital status x education and gender – for each item (Q! Q2 and Q3) it is exactly the same number (percentage).
  13. “3.2 Global analysis on environmental and behavioral factors” – there is no behavioral factors here.
  14. 2 – add the information if the difference was a relative difference or absolute one (in percent points).
  15. Correct “The samples were selected at random by gender….” – Do you mean Gender was not “strata” variable?
  16. Use the same name for MEHM – once it is “European Minimum Health Module”, in other place “Minimum European Health Module”.
  17. Correct the in-text citation format – there is no need to use two types of brackets - e.g. ([2]), ([4-5]), etc.
  18. Table A.1 - title should be changed – this is difference between men and women in age distribution (not distribution of participants).

Author Response

We thank you the Reviewer for this second, accurate revision. We hope we have satisfactorily addressed raised issues. Here attached you’ll find our point-by-point reply.
